# TSNet: Integrating Dental Position Prior and Symptoms for Tooth Segmentation from CBCT Images

**Linjie Tong**[*1]                                   linjie.19@intl.zju.edu.cn
**Jiaxiang Liu**[*1]                                  jiaxiang.21@intl.zju.edu.cn
**Yang Feng**[2]                                      yang0478@e.ntu.edu.sg
**Tianxiang Hu**[1]                                   tianxianghu@intl.zju.edu.cn
**Zuozhu Liu**[†1]                                    zuozhuliu@intl.zju.edu.cn
[1] *Zhejiang University*
[2] *Angelalign Inc.*

**Editors:** Accepted for publication at MIDL 2023

## Abstract

Automated dental diagnosis requires accurate segmentation of tooth from cone-beam computed tomography (CBCT) images. However, existing segmentation methods often overlook incorporating prior information and symptoms of teeth, which can cause unsatisfactory segmentation performance on teeth with symptoms. To this respect, we propose Tooth Symptom Network (TSNet), consisting of Dental Prior Guiding Data Augmentation (DPGDA) and Dental Symptom Shape Loss (DSSL), to improve segmentation performance for teeth with different clinical symptoms. Experiments show that TSNet outperforms all state-of-the-art methods across datasets with all kinds of symptoms with an average increase of 1.13% in Dice and 2.00% in IoU.

**Keywords:** Tooth symptoms, CBCT, Dental prior, Symptom shape loss

## 1. Introduction

Digital cone-beam computed tomography (CBCT) reconstruction has been shown to improve the effectiveness of dental treatment planning and management (Hao et al., 2022). A high-quality automated CBCT model reconstruction requires an accurate tooth segmentation from CBCT images (Weiss and Read-Fuller, 2019). Prior works are general approaches that were not specifically developed for tooth segmentation, and as a result may not attain perfect accuracy as well as robustness on tooth CBCT images (Ronneberger et al., 2015; Zhou et al., 2018; Valanarasu et al., 2021; Wang et al., 2022; Jain et al., 2021).

Figure 1(a) shows one normal tooth CBCT image and CBCT images with five symptoms (Decurcio et al., 2012; Fontenele et al., 2022; Liu et al., 2007; Hofmann et al., 2013; Kuo et al., 2016). Root canal therapy (RCT) leads to high-density filling imagery in the area of the Root canal, while filings lead to high density filling imagery and metal artifacts. The severity of metal artifacts differs between composite-metal and composite-resin. Multiply teeth usually appear in the middle of upper jaws, and their dental crowns are in the shape of small cones, while their roots are smaller than those of normal teeth. Permanent tooth

---

[*] Contributed equally
[†] Corresponding author

germ lacks normal dental parts, such as dental crowns. Prosthesis leads to high-density imagery and artifacts, which can be more severe if the material is metal. Since the number of images with these symptoms is significantly smaller than that of normal images, models do not have the chance to learn symptoms thoroughly, these degradations are likely to result in shape distortion of the tooth segmentation result as well as the misclassification of high-density filling imagery or artifacts as tooth when encountering certain symptoms.

Recognizing that the region surrounding the dental arch curve is not only the most informative area for tooth segmentation but also the region where various degradations occur, it is logical to emphasize the model's attention on this region of interest. In this paper, we propose a novel method named Tooth Symptom Network (TSNet), which consists of two designs. Firstly, Dental Prior Guiding Data Augmentation (DPGDA) incorporates tooth location information to prioritize the neighborhood of the dental arch curve, guiding the model's attention towards this crucial area. Secondly, Dental Symptom Shape Loss (DSSL) aims to minimize the disparity between the predicted tooth boundary and the ground truth, enabling the model to make more informed decisions when dealing with degraded images. By integrating two designs, semantic priors are embedded into the transformer layer, assisting in the extraction of semantic map (Jain et al., 2021), which can improve the performance of segmentation. Experimental results demonstrate the superior performance of TSNet in tooth segmentation on both normal images and images presenting diverse symptoms.

## 2. Method

We propose DPGDA which leverages the distribution of tooth positions to guide the sampling of CBCT images for data augmentation, as is illustrated in Figure 1(c). Figure 1(b) exhibits the framework of TSNet. To integrate prior information about the distribution of teeth, we initially extract position information from the dataset to generate a Dental Position Map, by dividing all CBCT images into $128 \times 128$ patches and recording the number of images that have teeth in each patch. Then, instead of using individual pixels, we employ patches (eg., $16 \times 16$ patches) to describe the sampling size. For each position, we calculate the sum of the corresponding patches in the Dental Position Map, resulting in the Dental Position Guiding Map. The map is then normalized to obtain a probability distribution that guides the sampling of CBCT images during data augmentation.

Besides, we propose DSSL to constrain the shape of tooth segmentation results, particularly in images presenting symptoms. Firstly, $I_p$ is obtained by softargmax (Chapelle and Wu, 2010) on the probability map that is outputted by the decoder. Then, the shape of the tooth is extracted in the ground truth $I_g$ and the prediction $I_p$. DSSL is defined as follows:

$$loss_{DSSL} = \frac{(shape(I_g) - shape(I_p))^2}{N}, \ \ shape(I) = 255 \cdot \frac{(I_x)^2 + (I_y)^2}{max\{(I_x)^2 + (I_y)^2\}}, \quad (1)$$

where $N$ denotes the number of pixels in the image, $I_x$ and $I_y$ denote the Sobel operator results in the x-direction and y-direction, respectively (Pratt, 2007).

## 3. Experiment

We construct a CBCT tooth image dataset comprising a training set and six test sets. The dataset consists of 160 patient samples with diverse symptoms, collected from hospitals in

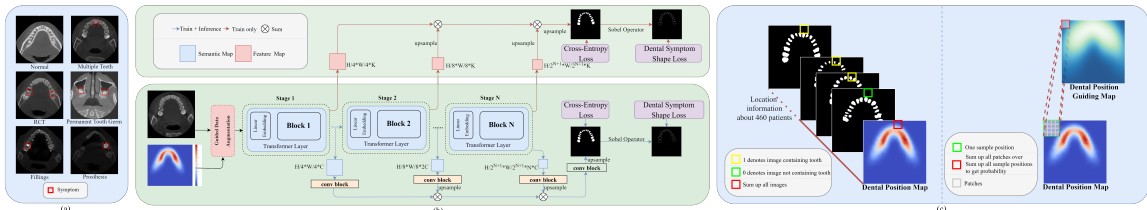

Figure 1: (a) includes typical six symptoms of tooth CBCT. (b) is the pipeline of TSNet. (c) is the process of DPGDA.

Table 1: Comparison of TSNet with state-of-the-art methods.

| | Normal | | RCT | | Fillings | | Multiply Teeth | | Permanent Tooth Germ | | Prosthesis | |
|---|---|---|---|---|---|---|---|---|---|---|---|---|
| | IoU | Dice | IoU | Dice | IoU | Dice | IoU | Dice | IoU | Dice | IoU | Dice |
| UNet | 84.04 | 91.33 | 80.19 | 89.01 | 84.28 | 91.47 | 83.87 | 91.23 | 85.95 | 92.45 | 76.02 | 86.38 |
| UNet++ | 84.25 | 91.45 | 77.5 | 87.32 | 83.75 | 91.16 | 83.24 | 90.85 | 85.42 | 92.14 | 69.79 | 82.21 |
| UCTransNet | 85.89 | 92.41 | 84.02 | 91.32 | 86.17 | 92.57 | 86.84 | 92.96 | 87.83 | 93.52 | 83.25 | 90.86 |
| MedT | 79.31 | 88.46 | 60.92 | 75.72 | 77.05 | 87.04 | 73.68 | 84.85 | 79.67 | 88.69 | 42.88 | 60.02 |
| SemaskT | 84.31 | 91.49 | 86.10 | 92.53 | 86.40 | 92.70 | 87.44 | 93.30 | 87.63 | 93.41 | 85.90 | 92.42 |
| TSNet | $86.92_{+1.03}$ | $93.00_{+0.59}$ | $87.81_{+1.71}$ | $93.51_{+0.98}$ | $89.22_{+2.82}$ | $94.30_{+1.60}$ | $89.05_{+1.61}$ | $94.21_{+0.91}$ | $90.22_{+2.39}$ | $94.86_{+1.34}$ | $88.31_{+2.41}$ | $93.79_{+1.37}$ |

China during 2018-2021. The training set consists of 1906 images from 100 **normal** individuals. The six test sets encompasses 196 images from 10 **normal** patients, 198 images from 10 **RCT** patients, 189 images from 10 **fillings** patients, 184 images from 10 **multiply teeth** patients, 97 images from 10 **permanent tooth germ** patients, and 203 images from 10 **prosthesis** patients. To evaluate the effectiveness of TSNet, we compare it with five other methods: U-Net (Ronneberger et al., 2015), U-Net++ (Zhou et al., 2018), MedT (Valanarasu et al., 2021), UCTransnet (Wang et al., 2022), and Semask T (Jain et al., 2021). To ensure fairness, all baseline methods were implemented using the original source code, with default hyperparameters and evaluation metrics were computed using MMsegmentation (Contributors, 2020). Experimental results, as shown in Table 1, demonstrate that TSNet outperforms the other methods across all datasets.

## 4. Conclusion

In this work, we design two modules, DPGDA and DSSL, for tooth segmentation tasks, and propose an integrated tooth segmentation method called TSNet. We evaluate TSNet's performance by comparing it with five advanced image segmentation methods on six datasets containing normal tooth CBCT images and images with different symptoms, TSNet demonstrates superior segmentation performance. Moving forward, we anticipate the application of TSNet in clinical settings to aid in the diagnosis and treatment for dental diseases.

## Acknowledgments

This work is supported by the National Natural Science Foundation of China (Grant No. 62106222), the Natural Science Foundation of Zhejiang Province, China (Grant No. LZ23F020008), the Scientific Research Fund of Zhejiang University (XY2022025), and the Zhejiang University-Angelalign Inc. R&D Center for Intelligent Healthcare.

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
