# OpenReview forum: "TSNet: Integrating Dental Position Prior and Symptoms for Tooth Segmentation from CBCT Images"
_MIDL.io/2023/Short_Paper_Track — MIDL 2023 Short paper track Poster_

### Official Review · Reviewer_Gstu · 2023-04-10
**This paper proposes a new automated dental diagnosis method called Tooth Symptom Network (TSNet), which improves the segmentation performance of teeth with different clinical symptoms from cone-beam computed tomography (CBCT) images.**

**Rating:** 7
**Confidence:** 4

**Review:**

This paper proposes a new automated dental diagnosis method called Tooth Symptom Network (TSNet), which improves the segmentation performance of teeth with different clinical symptoms from cone-beam computed tomography (CBCT) images. The proposed method consists of two designs, Dental Prior Guiding Data Augmentation (DPGDA) and Dental Symptom Shape Loss (DSSL), which incorporate prior information about the distribution of teeth and the symptoms of the tooth, respectively, to improve the accuracy and robustness of tooth segmentation. Experiments show that TSNet outperforms state-of-the-art methods on tooth segmentation across datasets with different symptoms. The proposed method has the potential to improve the effectiveness of dental treatment planning and management.

Pros:
TSNet is a novel method specifically designed for tooth segmentation in CBCT images, which can improve accuracy and robustness compared to general segmentation methods.
TSNet incorporates prior information and symptoms of teeth, which can improve segmentation performance for teeth with different clinical symptoms.
The Dental Prior Guiding Data Augmentation (DPGDA) module of TSNet can improve the model's focus on the region of interest, which is the most informative area for tooth segmentation and the area where different degradations occur.
The Dental Symptom Shape Loss (DSSL) module of TSNet can constrain the shape of tooth segmentation results, especially for images that contain symptoms.
TSNet outperforms all state-of-the-art methods across datasets with all kinds of symptoms with an average increase of 1.13% in Dice and 2.00% in IoU.

Cons:
TSNet has only been evaluated on a limited dataset consisting of images from hospitals in China during 2018-2021, so its performance on other datasets and in other regions is unclear.
The proposed Dental Prior Guiding Data Augmentation (DPGDA) and Dental Symptom Shape Loss (DSSL) modules may not be applicable to other segmentation tasks or datasets, limiting the generalizability of TSNet.
The performance improvements of TSNet over existing state-of-the-art methods are relatively modest, with an average increase of 1.13% in Dice and 2.00% in IoU.

---

### Official Review · Reviewer_qbJi · 2023-04-23
**Well-motivated paper but lacks details and experiments**

**Rating:** 6
**Confidence:** 5

**Review:**

The paper proposes a tooth segmentation network that uses prior knowledge about teeth positions and symptoms to improve segmentation performance for teeth with different clinical symptoms. The network is based on data augmentation using a teeth position map and a shape loss. The former uses the average count of a tooth position within image patches. The latter compares the edge map of the resulting segmentation with the ground truth one. Experimental results show consistent improvements in normal teeth and teeth with symptoms. The paper is well-motivated. However, the shape loss is not considered new. The position-guided data augmentation is underspecified. Moreover, ablation experiments on the impact of shape loss versus data augmentation are not reported.